# Short-lived calcium carbonate precursors observed in situ via Bullet-dynamic nuclear polarization
Ertan Turhan[1,2], Masoud Minaei[3], Pooja Narwal [3], Benno Meier [3,4] ✉, Karel Kouřil [3] &
Dennis Kurzbach [1] ✉

The discovery of (meta)stable pre-nucleation species (PNS) challenges the established nucleation-and-growth paradigm. While stable PNS with long lifetimes are readily accessible experimentally, identifying and characterizing early-stage intermediates with short lifetimes remains challenging. We demonstrate that species with lifetimes ≪ 5 s can be characterized by nuclear magnetic resonance spectroscopy when boosted by 'Bullet' dynamic nuclear polarization (Bullet-DNP). We investigate the previously elusive early-stage prenucleation of calcium carbonates in the highly supersaturated concentration regime, characterizing species that form within milliseconds after the encounter of calcium and carbonate ions and show that ionic pre-nucleation species not only govern the solidification of calcium carbonates at weak oversaturation but also initiate rapid precipitation events at high concentrations. Such, we report a transient co-existence of two PNS with distinct molecular sizes and different compositions. This methodological advance may open new possibilities for studying and exploiting carbonate-based material formation in unexplored parts of the phase space.

The observation of metastable pre-nucleation clusters (PNC) that precede the formation of many solid phases and seemingly contradict the classic nucleation-and-growth paradigm has led to a *renaissance* of crystallization research in the past decade[1–3]. PNC have been reported for a wide range of solidification mechanisms, including calcium phosphates[2,4–9] and carbonates[10–16]—two major inorganic components of biological solid materials. These studies are important due to their fundamental character for understanding material and biomaterial formation, as well as due to the possibility of rationalizing material design by controlling the precipitation pathways *via* early-stage precursors.

PNC are thought of as (meta-)stable ionic assemblies, which form in solution and represent the first step in the formation pathway of many solid ceramics. Studies of PNC properties under varying experimental conditions report a coalescence and dehydration of PNC that leads to the formation of solid phases. Such solidification mechanisms challenge the classical nucleation-and-growth theory (CNT)[16–18]. Yet, it is still debated whether the observation of PNC can be reconciled with the CNT. The discussion is hampered by scarce experimental evidence, not least due to the highly dynamic character of PNC.

However, almost all studies have focused so far on a concentration regime of mild oversaturation[1,3,16,19,20]. Under such conditions, the observation of PNC is experimentally feasible due to their (meta)stability and long lifetimes[20–25]. However, higher concentration regimes, where the early-stage precursors feature lifetimes of only a few seconds, remain very challenging to access and almost unstudied. Indeed, identifying and characterizing such species under high oversaturation conditions requires novel experimental approaches[3]. Developing such methods is yet a challenge worthy of tackling as the untapped concentration regime is large and may enclose many undocumented precipitation pathways, which may help to shed light on non-classical crystallization phenomena. Indeed, precursor species need to be identified and characterized to further our understanding of solidification mechanisms and find routes toward controlling material formation pathways.

Recently, it was proposed to employ nuclear magnetic resonance (NMR) boosted by dissolution dynamic nuclear polarization (DDNP) to overcome the experimental challenge of accessing precursor species in the high-oversaturation regime, which were denoted as prenucleation species (PNS)[26,27]. (Note that we distinguish the notion of prenucleation cluster,

[1]Institute of Biological Chemistry, Faculty of Chemistry, University of Vienna, Währinger Str. 38, 1090 Vienna, Austria. [2]University of Vienna, Vienna Doctoral School in Chemistry (DoSChem), Währinger Str. 42, 1090 Vienna, Austria. [3]Institute of Biological Interfaces 4, Karlsruhe Institute of Technology, 76344 Egenstein-Leopoldshafen, Germany. [4]Institute of Physical Chemistry, Karlsruhe Institute of Technology, 76131 Karlsruhe, Germany. ✉e-mail: benno.meier@kit.edu; dennis.kurzbach@univie.ac.at

PNC, from PNS. While dynamic and energetic properties define the former[28], the latter is herein thought as a generic description of material precursor species, not invoking any particular definitions.)

With DDNP[29–31], NMR resonances can be boosted over 10.000-fold, enabling signal acquisition in less than a second. When coupled with in-situ mixing techniques[32–34], the detection of very early-stage precursors becomes possible within less than a second after sample preparation, hence allowing access to short-lived PNS encountered under high oversaturation. This was demonstrated for calcium phosphates exploiting the $^{31}P$ resonance of the phosphate ions[26,27].

The highly relevant system of calcium carbonate (CaC)[15,35,36], however, is not accessible by conventional DDNP experiments except at elevated pH values above 10. In such an experiment[37], the nucleus of interest is hyperpolarized, *i.e.*, signal-enhanced ex-situ in a dedicated DNP apparatus at cryogenic temperatures[31]. After completion of the hyperpolarization procedure, the sample is dissolved, and the resulting solution is pushed, typically under ca. 3–7 bar overpressure, through a capillary to an NMR spectrometer for detection[29]. The high chase gas pressure (helium gas) leads to a degassing of the aqueous hyperpolarized solution[38]. Hence, carbonate is effectively purged from the sample in the form of $CO_2$ during transfer to the NMR spectrometer when experimenting at near-physiological pH values.

Herein, we show that this shortcoming of DDNP, *i.e.*, its inapplicability to pressure-sensitive samples, can be overcome with so-called 'Bullet'-DNP[39,40]. In a Bullet-DNP experiment, the sample is transferred as a frozen pellet to the detection spectrometer and only dissolved immediately before NMR detection. Hence, the sample experiences the chase gas pressure only in the frozen state and retains its carbonate content during transfer and dissolution. Using this method, we demonstrate that multiple short-lived (lifetimes $\ll 5$ s) CaC precursors[12] can be simultaneously identified and characterized, which form immediately after mixing a hyperpolarized carbonate solution with a calcium chloride solution. We thus demonstrate that the very early-stage CaC precursors in the high oversaturation regime are accessible and characterizable by NMR spectroscopy, evidencing the existence of non-classical crystallization pathways in this formerly untapped part of the CaC phase space.

## Results and discussion

For Bullet-DNP, we followed a protocol based on the procedure detailed in references[39,40]. In brief, 50 µL of a 500 mM solution of $^{13}C$-enriched sodium carbonate solution was hyperpolarized at a temperature of $T_{DNP} = 1.5$ K using the OX063 radical as a polarization agent. Upon completion of the signal build-up, the frozen sample was shot to the NMR spectrometer ($B_{0,NMR} = 9.4$ T, $T_{NMR} = 298$ K) into a newly developed injector, where a brass stopper caught the sample cup. This led to an ejection of the frozen sample directly into an NMR tube, where it was dissolved in 500 µL of 20 mM calcium chloride in 50 mM HEPES, corresponding to a >200-fold oversaturation at pH 6[41]. These conditions were chosen to probe so far unstudied parts of the phase space, while being well compatible with experimental Bullet-DNP conditions. The injection and mixing system is detailed in Fig. 1. More details on the Bullet-DNP method can be found in reference[39].

Directly upon mixing, we observed three resonances. That of free carbonate at $\delta(^{13}C) = 160.4$ ppm (see Fig. S1 and ref. 42 for the resonance assignment) next to that of two further $^{13}C$ resonances centered around $\delta(^{13}C) = 162.3$ ppm and at $\delta(^{13}C) = 168.9$ ppm (Fig. 2). In addition, a weak resonance of $CO_2$ could be observed at 125.9 ppm (see the Supporting Information Fig. S2). Note that bicarbonate ($HCO_3^{2-}$) species in solid amorphous CaC also display a chemical shift of 162 ppm[24]. Hence, the PNS species at 161.4 ppm may be primarily bicarbonate-based. However, neither the relative abundances of $H_2CO_3$, $HCO_3^-$, and $CO_3^{2-}$ are known during PNS formation, nor are the proton exchange rates. The exchange between $H_2CO_3$ and $HCO_3^-$ is very fast for neat sodium carbonate in solution, though, and hence, it likely remains fast in our experiment, too. The possibility that the species at 161.4 ppm is pure bicarbonate is therefore discarded since bicarbonate would always be in fast exchange with other carbonates.

The data presented in Fig. 2 are corrected for convection by using the signal of hyperpolarized glycerol as an internal standard (see Supporting Information Fig. S3). Immediately after injection, the detected signal intensity of glycerol (and hence carbonate) is low due to physical sample movement. The correction procedure, therefore, increases both the signal and the noise (details of the procedure are given in the Supporting Information). Notably, strong convection implies that substrate polarization cannot be derived directly from the signal-to-noise ratio.

Two spectral features can be straightforwardly exploited to understand the second species in the Bullet-DNP experiments: line shapes and signal intensities.

The linewidth of the free carbonate at 160.4 ppm was $9 \pm 1$ Hz (for the assignment of the chemical shift of this species, see Fig. S6), that of the second signal at 161.4 ppm was $18 \pm 1$ Hz, and that of the strongly broadened species centered around 168.9 ppm was $318 \pm 21$ Hz. Similar broadened resonances in the context of DDNP experiments on calcium phosphate (CaP) precipitation have recently been associated with slow-tumbling calcium phosphate early-stage precursors[26,27]. In this interpretation, the ions are confined in larger clusters, leading to slower molecular tumbling and correspondingly higher $R_2$ rates, which in turn leads to a larger linewidth $\Gamma$. Hence, for the case at hand, CaC clusters, *i.e.*, PNS are the most likely species underlying the detected broad lines.

Considering that the hydrodynamic radius of bicarbonate[43] is ca. 0.2 nm and following the framework established in reference[1], an increase in linewidth by a factor of $318/9 \approx 35$ corresponds to an at a most 35-fold increase in rotational correlation time and, thus, a <3.3-fold increase in hydrodynamic radius. However, it should be noted that the linewidth $\Gamma$ can also be influenced by exchange processes, such that the 3.3-fold increase must be considered as a theoretical maximum.

Interestingly, for the presented case, we observed two additional peaks, well-separated from that of the free carbonate, with significantly different line widths. Following the interpretation outlined above, this points towards two different CaC precursor species. One consists of a smaller PNS housing a small number of carbonates. The second consists of a larger ionic assembly with significantly reduced rotational mobility relative to the free carbonate species.

This observation aligns very well with the reported CaC non-classical precipitation pathway[25] in which smaller CaC species form, typically ion pairs (corresponding to the signal at 162 ppm), that assemble in a secondary event into larger assemblies or condensed liquid phases (corresponding to the signal at 169 ppm)[20]. Hence, our observations indicate that the material formation mechanism observed under mild oversaturation might also be relevant under strong oversaturation, albeit featuring much faster kinetics.

Qualitatively, higher degrees of oversaturation entail shorter lifetimes, as well as larger observed species, as reflected in broader lines[26]. However, it should be noted that the faster kinetics at high concentrations and the associated acceleration of the solidification event also imply that parts of the precipitation process may take place outside the time window accessible to bullet-DNP.

In an alternative scenario, the species resonating at 161 ppm does not transform into CaC. Instead, free carbonate directly transforms into PNS without any intermediate step, and the second species forms without taking part in any further event. At present, however, our data cannot precisely reveal the underlying pathway. Since the concentrations change rapidly during signal acquisition, it is also possible that the system wanders from one part of the phase space to another, such that multiple CaC formation pathways might be relevant. Nevertheless, our observation can generally be reconciled with the existing models, which show that CaC precipitation can cover several precursor species[16,20,44]. The chemical shift difference between the smaller and the larger PNS is >6 ppm. Hence, the two species also differ in local structures, specifically the chemical environment of the $^{13}C$ nuclei. For example, local pH, ion coordination, hydration status, and exchange kinetics between different sites might differ.

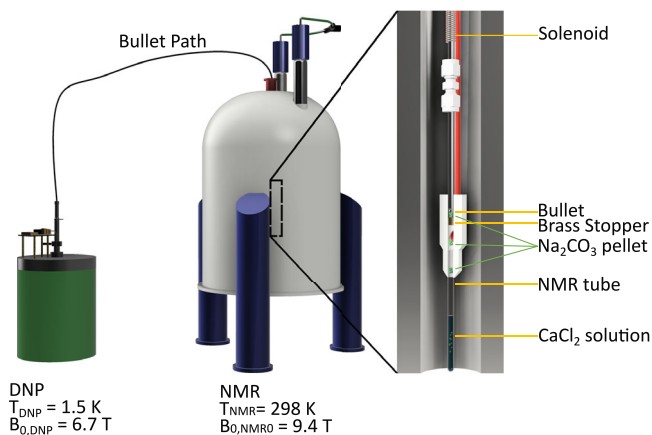

**Fig. 1 | Sketch of the experimental setup.** A carbonate solution is pipetted into a miniature bucket referred to as bullet. The material is frozen in liquid nitrogen and the bullet is loaded into the polarizer, where it is hyperpolarized at cryogenic temperatures using slightly off-resonance microwave irradiation. The hyperpolarized pellet is then pneumatically 'shot' to an NMR spectrometer operating at room temperature. There, the bullet itself is retained in a constriction (brass stopper). The frozen solution is catapulted out of the bullet and, thus, eventually dissolved through the force of impact on the calcium chloride solution surface. Immediately after the dissolution and mixing event, NMR detection is triggered. Through this custom setup, the carbonate sample does not experience any high temperatures or pressures during transfer and dissolution. At the same time, very rapid molecular interaction mechanisms can be resolved.

Earlier work classified mineral precursors into type 1 and 2 PNS[3,26]. The former are stable (or metastable) in solution and only initiate the solidification process upon external stimuli, such as pH or temperature jumps. The latter appears as a transient species in precipitation events, which form the final solid without any further trigger (although this definition does not exclude longer-lived steady-state phases[45]). For the PNS reported herein, evidently, the latter case applies, *i.e.*, the observed PNS are immediately transformed into a solid without the need for any further stimuli.

To confirm our data and data interpretation, the experiments were repeated at pH 7 and pH 8. In both cases, a similar behavior was observed (see the Supporting Information Figs. S5–S6).

The intensities of the resonances depend on the relative abundance of the species and their conversion kinetics. Figure 3 shows the time dependencies of the relative signal intensities for the three observed species. These traces decay with an effective relaxation rate $R_{eff}$, which is a combination of the longitudinal $^{13}C$ relaxation rate and the conversion of free carbonate into CaC precursor species followed by the formation of solid CaC.

For the free carbonate, we found an $R_{eff}$ of $0.08 \pm 0.01 \, s^{-1}$. For the other two $Ca^{2+}$-bound species, we found $0.37 \pm 0.03 \, s^{-1}$ and $0.50 \pm 0.03 \, s^{-1}$, respectively. The free species show a significantly slower relaxation rate than the $Ca^{2+}$-bound ones, which might suggest that all (or most) of the $Ca^{2+}$-ions are rapidly bound in a PNS upon mixing, such that the free carbonate species is not affected by the CaC precipitation event.

Experiments conducted at pH 6, 7, and 8 yielded relative ratios of $0.62 \pm 0.2$, $0.87 \pm 0.3$, and $0.96 \pm 0.4$, respectively, between the broad and narrow PNS resonances as measured 2 s after the start of signal acquisition (derived from the ratio of signal integrals; see the Supporting Information Fig. S7). Hence, higher pH led to faster conversion of the small ionic clusters into larger aggregates/condensed phases. Five seconds after mixing, the broad species had already mostly disappeared in all three cases. Hence, even despite very rapid formation mechanisms, Bullet-DNP could determine the relative pace of CaC early-stage conversion.

The result of the completed solidification process can be probed using powder X-ray diffraction. This yielded pH-dependent contributions of vaterite and calcite (see the Supporting Information Fig. S8). Hence, the observed PNS did not lead to one specific solid.

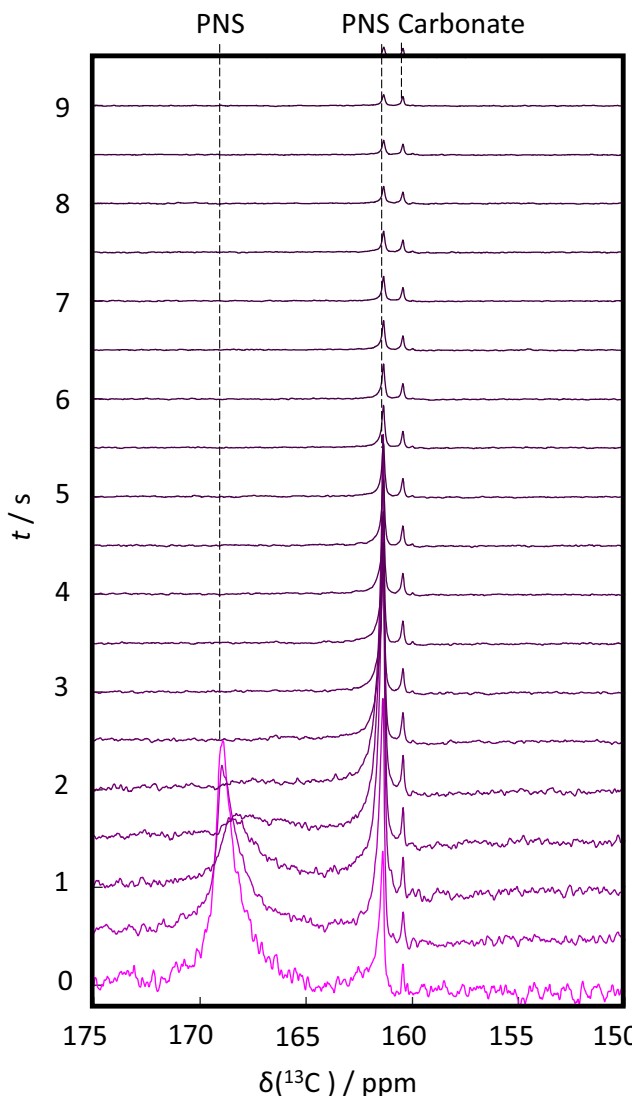

**Fig. 2 | Hyperpolarized time-resolved 13 C NMR spectra.** Time evolution of hyperpolarized $^{13}C$ NMR spectra obtained after mixing the calcium chloride and carbonate solutions at pH 6. Note that the noise level is changing because of the data correction procedure (see the Supporting Information Fig. S4).

The rapid sample precipitation also complicated the determination of the solution-state NMR signal enhancement factors ε. Typically, in DDNP applications, the signal intensity in thermal equilibrium, *i.e.*, after decay of the hyperpolarization, is used as a reference. Evidently, since the hyperpolarized substrate precipitates rapidly from the solution, such an analysis becomes unfeasible. We were nevertheless able to record a reference spectrum for the free carbonate and the narrow PNS resonance at pH 6 in thermal equilibrium after the completion of the CaC formation event. This is compared to a representative spectrum recorded 2 s after mixing in Fig. 4 and led to an apparent signal enhancement of $ε = 52.700 \pm 500$.

## Methodological Considerations

Two aspects need to be considered when applying bullet-DNP as a method to assess very fast material formation processes.

Firstly, gas bubbles may lead to substantial line broadening in D-DNP experiments. However, such bubbles were suppressed effectively by degassing the solvent and by filling the NMR tube with a sufficient volume to ensure magnetic field homogeneity across the sensitive region of the NMR detection coil after the impact of the sample on the solvent. The observation

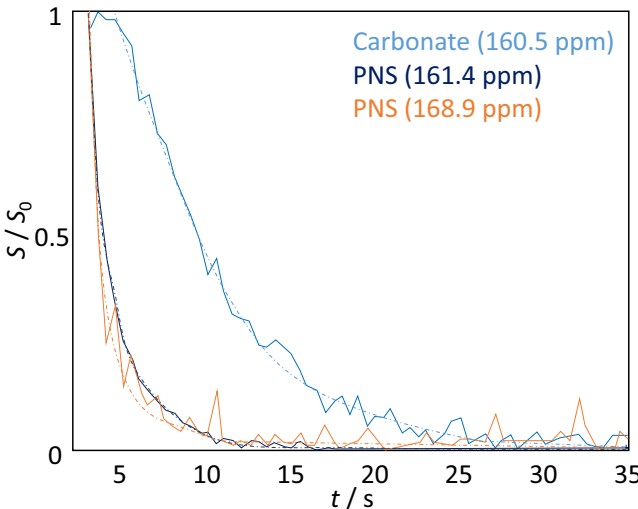

**Fig. 3 | Evolution of signal intensities.** Signal intensities and fits to exponential decay functions of the three recorded signals as a function of time at a pH of 6. $S/S_0$ denotes signal intensity, which means that the data is normalized to the first data point shown in the figure. For the non-normalized data, see the Supporting Information Fig. S10. In the figure, $t = 0$ corresponds to the time of mixing.

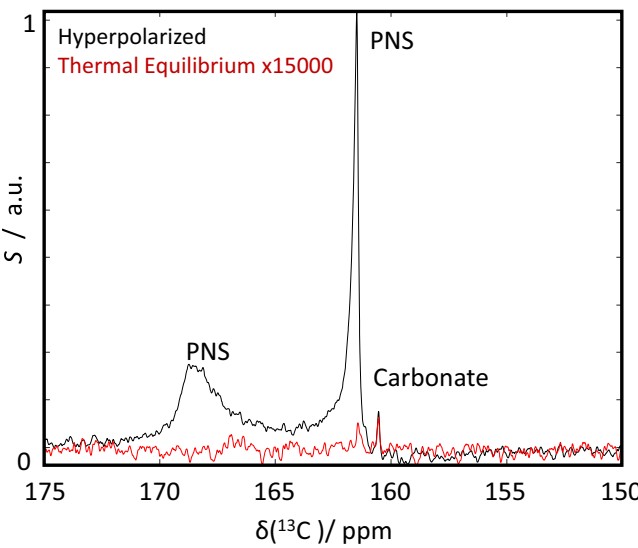

**Fig. 4 | Comparison of hyperpolarized and thermal equilibrium spectra.** Representative spectrum recorded 1 s after the start of the acquisition (black) and thermal equilibrium reference (red, average of 512 scans). The differences in line shapes of carbonate and the two PNS is evident in the hyperpolarized spectrum.

of the narrow resonance of the free carbonate is strong evidence for a genuine broadening mechanism.

Furthermore, the line shapes were not distorted by convection or shim problems as shown by the narrow $^{13}C$ linewidth (9 Hz) of the admixed glycerol-$d_8$ (see the Supporting Information)

Secondly, the solution present immediately after the dissolution of the bullet is likely not fully homogenous, and it might take several seconds for the sample to homogenize in terms of concentrations. Hence, the degree of oversaturation might also vary across the sample space. Therefore, it is important to use appropriate correction functions (see Supporting Information) to derive signal intensities. The reported kinetic data potentially remain biased to some degree by concentration gradients as the reaction rate might change. The presence of glycerol may likewise impact the CaC formation.

Reproducing the exact same experiments with two different "traditional" DDNP systems, where the sample is dissolved in the DNP magnet (see[46,47] for the details on the used DDNP systems) and then pneumatically propelled by a chase gas to the detection NMR system did lead to a complete purge of carbonate from the solution (see the Supporting Information Fig. S9). It should be noted that Balodis et al. showed that DDNP observations of calcium carbonate precursors are possible at elevated pH values of 10 or higher[37].

## Conclusions

We draw two major conclusions from the reported results. Firstly, using Bullet-DNP, it is possible to record spectra enhanced by over four orders of magnitude even for very temperature or pressure-sensitive samples. This feature widens the scope of ex-situ hyperpolarization techniques. For the case of hyperpolarized carbonate, the first seconds of $Ca^{2+}$-bound precipitation could thus be accessed under high oversaturation by $^{13}C$-NMR, which remained out of the scope of traditional DDNP or any other high-resolution method.

Secondly, we showed that pre-nucleation species can be identified and characterized by NMR immediately after exposing calcium and carbonate ions to each other. The presence of these PNS signals can be explained by a non-classical precursor pathway—as reported for mild oversaturation[12,20,28]— responsible for CaC formation also under high oversaturation conditions.

It should be noted that earlier work on calcium phosphates using DDNP[26,27] to monitor prenucleation species led to comparable observations, *i.e.*, concerted detection of PNS and free phosphate ions in solution. However, as calcium phosphates feature a much lower solubility than carbonates, these studies were performed at lower degrees of oversaturation. More importantly, though, the main difference is that 'traditional' DDNP could be applied as phosphate ions remain in solution even under high chase gas pressures. In combination, DDNP and Bullet-DNP provide means to study prenucleation phenomena for a wide range of target systems, including biologically most important oxyanions[19,27,28,48].

Given the high interest in deciphering the structure-function relationship of CaC pre-nucleation species due to their potential to control the morphologies of mineral phases[12–16,19,24,25,28,35,44,49], the presented methodology enables the application of NMR, a versatile high-resolution structure-determination tool, to the ongoing endeavors in non-classical nucleation research. In particular, applications of quantification methods for interconversion, such as ultrafast exchange and diffusion spectroscopy, might become possible.

Since Bullet-DNP is sensitive to very early-stage precursors of CaC at relatively high ion strengths, a combination with molecular dynamics (MD) simulations appears worthwhile. Not only are MD simulations applicable to processes taking place on the microseconds time scale (or milliseconds when coarse-graining), but the high concentrations also help to reduce computational costs, as box sizes can be reduced.

The reported methodology can widen the accessible precursor space in the design of CaC-based functional materials. In particular, endeavors that aim to predetermine the morphology and structure of a carbonate-based solid by choosing a particular PNS as precursor might profit from the reported methodology. The solidification pathways from a PNS to a final material in the almost untapped high oversaturation regime of the CaC phase space may lead to new morphologies with unexpected properties.

## Methods

Sample Preparation: 0.1 M HEPES (2-[4-(2-hydroxyethyl)piperazin-1-yl] ethanesulfonic acid) were dissolved in 40 mL of water. Subsequently, the pH was adjusted to 6, 7, or 8. Sonication under vacuum for 15 min ensured the removal of dissolved oxygen. The $CaCl_2$ solution was prepared by dissolving 4.6 mg of $CaCl_2$ in 1 mL of the buffer ($c_{CaCl_2}$ = 0.041 M). The sample was prepared for Bullet-DNP by transferring 600 µL of the obtained solution with an additional 50 µL of $D_2O$ into an NMR tube. This yielded a $CaCl_2$ solution with a concentration of 0.038 M.

For DNP, 0.5 M of $^{13}$C labeled $Na_2CO_3$ were dissolved in a mixture of glycerol-$d_8$/$D_2O$ and HEPES buffer at pH 7 in a volumetric ratio of 50:40:10. The OX063 radical was used as polarization agent. It was co-dissolved in the resulting solution at a concentration of 0.015 M. For each subsequent DNP experiment, 50 µL of this stock solution was utilized. The solution was filled into a bullet (see reference[39]), frozen with liquid nitrogen, and inserted into the DNP system. DNP was performed using the system described in reference[46] at a temperature of 1.5 K and a magnetic field of 6.7 T until a build-up saturation was observed. The best build-up kinetics were observed using continuous microwave irradiation at 187.650 GHz. To maintain the hyperpolarization during the transfer of the hyperpolarized bullet, the passage was sheltered using pulsed solenoids that provided a magnetic field of 70 mT. The hyperpolarized sample was dissolved in the NMR tube in the $CaCl_2$ solution through the force of impact. The resulting sample volume was 700 µL. This procedure resulted in a final carbonate concentration of 35.7 mM and a final $CaCl_2$ concentration of 35.1 mM.

For detection, decoupling was not applied, and the probe temperature was set to 298 K.

All data were zero-filled to twice the FID lengths and apodized using an exponential window function, adding a 5 Hz line broadening before Fourier transformation (FT). After FT, all data were baseline corrected. For analysis of signal amplitudes, all peaks were then fitted using the 'fitnlorentzian.m' function for MATLAB 2022. The resulting signal amplitudes were then corrected for sample convection using the solvent signals (see the Supporting Information for the detailed procedure).

## Data availability
All data and codes are available under https://phaidra.univie.ac.at/o: 2082347.

## Code availability
All data and codes are available under https://phaidra.univie.ac.at/o: 2082347.

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

## Acknowledgements

The authors acknowledge support from the NMR core facility of the Faculty of Chemistry, University of Vienna. The project received funding from the European Research Council (ERC) under the European Union's Horizon 2020 research and innovation program (Grant agreements 801936 and 951459), from the "Impuls- und Vernetzungsfonds of the Helmholtz-Association" under grant VH-NG-1432 and from the Austrian FWF (project numbers P33338-N and I5771-N).

## Author contributions

E.T., M.M., P.N., B.M., K.K., and D.K. performed the experiments. B.M. and D.K. designed the study. E.T., B.M., and D.K. wrote the manuscript. All authors read and approved the manuscript.

## Competing interests

The authors declare no conflict of interest.
