## [Peer Review File · Communications Chemistry]

Reviewers' comments:

Reviewer #1 (Remarks to the Author):

Authors of the manuscript "Bullet-DNP enables in-situ observation of multiple short-lived calcium carbonate precursors – non-classical crystallization under strong oversaturation" described the process of in situ observation of multiple short-lived precursors of calcium carbonate–non-classical crystallization under strong supersaturation. Turhan et al. demonstrates how Bullet-DNP was used to investigate the initial stages of calcium carbonate precipitation under high supersaturation conditions. The authors further demonstrate how Bullet-DNP, which involves the transfer of a hyperpolarized frozen sample directly into the NMR spectrometer, makes it possible to detect several short-lived CaCO₃ PNS, whose lifetime is less than 5 seconds.

The manuscript is well structured and interesting to read. Here are some optional comments to address:

- What are the next steps to further elucidate the atomic-level structure of the short-lived PNS using Bullet-DNP in combination with other structural techniques?
- How does the degree of supersaturation quantitatively affect the lifetimes of the different PNS observed? and sizes of PNS?

Reviewer #2 (Remarks to the Author):

Meier, Kurzbach and coworkers report an interesting application of "bullet dDNP", a hyperpolarising liquid NMR technique developed a few years ago, to the investigation of the early stages of the formation of calcium carbonate. This application brings an important advantage as it allows to overcome current limitations of traditional dDNP approaches for the study of pressure sensitive samples, so it has a potentially large application to study fast events in these systems. The manuscript is concise and clear, the references appropriate, and the results are potentially interesting.

However this study lacks strong evidence for their conclusions, as I see a problem with the NMR data interpretation.

My main criticism on this study is related to the assignment of the two signals observed at 162 and 169 ppm (shown in figure 2) to clusters of different sizes.

In the manuscript, the signal with smaller linewidth (at 162 ppm) is assigned to smaller calcium carbonate clusters, while the one with the larger linewidth (169 ppm) to larger calcium carbonate clusters. While this interpretation is purely based on dynamics (larger clusters should be slower than smaller ones, which is understandable), I was surprised that isotropic chemical shifts are not discussed at all in the manuscript. In particular, it seems important to discuss why two carbonate species that only differ in their size should show such a large (more than 6 ppm) chemical shift difference?

It is claimed that this interpretation aligns well with previous articles on the formation of prenucleation clusters of calcium carbonate, with a citation to ref. 18 (Demichelis et al.). However, an important aspect of Ref 18 is the discussion of the role of bicarbonate ions in carbonate cluster speciation, at different pH. Now, in the present manuscript, there is no mention at all of the possible presence of bicarbonate ions. The (computational) study in Ref 18 was recently used to support interpretation of NMR data reported in another article from some of the same authors (cited as ref 22 in this manuscript). Ref 22 clearly shows that ¹³C signals of bicarbonate ions in solid amorphous calcium carbonates appear at ~162 ppm, while signals of calcium carbonate in amorphous environments are expected at ~168 ppm.

Therefore, I strongly suspect that the assignment discussed in this manuscript is not correct. It is important that data interpretation is revised in light of the cited studies before the manuscript becomes acceptable for publication in this Journal.

Related to this point: concerning the discussion on the decay curves at different pHs ("higher pH led to faster conversion of the small ionic clusters into larger aggregates/condensed phases"), the observed behaviour could be justified, too, by considering the different assignment of the two observed peaks to calcium carbonate in prenucleation species (169 ppm) and bicarbonate ions (162 ppm), as, with increasing the pH, the equilibrium between bicarbonate and carbonate ions

should be displaced more towards the carbonate ions.

Other minor points:

-The manuscript claims that that standard dDNP techniques are incompatible with the detection of calcium carbonate clusters because of the degassing step. However, no proof is provided or previous reference. It would be instructive to see these data added to the SI.

-It is not clear why such high supersaturation levels were employed in this study. Why are these conditions relevant? This point should be mentioned in the revised manuscript.

Reviewer #3 (Remarks to the Author):

The work of Turhan et al. uses a specific and novel form of hyperpolarized solution NMR spectroscopy where the sample is shuttled rapidly from the hyperpolarizer to the NMR magnet, where it is dissolved for study. In this case, this enables the signal-enhanced time-resolved study of product nucleation during a solution reaction between sodium carbonate and calcium chloride. Interestingly, different carbonate species are observed after only a few seconds of mixing. This is clearly an important technique that will help researchers understand the first steps of crystallisation. However, there are some remaining questions that need to be addressed before this work can be published. These are provided below:

1. What polymorph(s) of CaC are precipitated from this reaction? The pXRD of the precipitate should be shown. Do the PNS lead to one polymorph, or two (or more)? Does this vary with pH? If so, can this be linked to the particular PNS as the larger clusters are more prevalent at higher pH?
2. What role does glycerol play in NC formation? This should be discussed in the main text and not only in the SI. In the SI the authors state that glycerol can be assumed to not take part in CaC formation, with a citation, and yet the cited paper (by some of the authors) states: "It should be mentioned that the D-DNP approach employs additives such as glycerol and the polarizing agent (TEMPOL). These specific conditions might influence the precipitation process, which therefore cannot directly be compared to previously published data on CaP systems." So there is a contradiction. It could well be expected that glycerol plays a role in stabilizing clusters as a ligand, as water would also be expected to do. The experiments should be repeated using DMSO (or equivalent cryoprotectant) to ensure the same species are observed. The sodium carbonate may help pure water form a glass and thus no cryoprotectant may be required for DNP. It is worth trying.
3. State the full name of HEPES.
4. Fig. S6 missing.
5. What's happening in Fig. S1? There is not sufficient information in the caption to understand where this exemplary spectrum comes from – and it's labelled as exemplary but does not resemble others (e.g. Fig. 2). Why is the peak near 169 ppm so much larger than in Fig. 2?
6. Why is the noise larger at shorter times (Fig. 2, S3 (left), but not S3 (right))? Yet it's the same in S4. This should be explained.
7. Fig. 3 does not look like it matches the data. Also, the caption should specify which pH sample. Are they relative to carbonate intensity? If so, this wouldn't be S/S0. How is S/S0 above 1? This should be clarified in the caption. For the 169 ppm resonance, it is almost gone at 3 s (as stated below in the text), whereas the 161 ppm resonance remains and appears to have a different decay constant in Fig. 2, whereas in Fig. 3 it appears the same for the 169 and 161 species. Something is amiss.
8. Main text has integral ratios measured after 1 s, whereas caption to Fig. S7 has 2 s. Also, integral ratios should not be measured in %. An integral "ratio" of 96% could suggest a ~1:1 ratio.
9. Why narrower for the 169 ppm peak at pH 7? Same species but small cluster? Does this infer there are no magic cluster sizes here?
10. Why is the black spectrum in Fig. 4 different to the blue spectrum in Fig. S7 when they have been recorded under the same conditions? Should they not be the same spectrum? How many repeat measurements were done? What is then the error on the integral ratios? It looks very large comparing Figs. 4 and S7 for the pH 6 sample.
11. Typo: "However, it must be considered that the kinetic data shown in Fig. 4 potentially

remains" should be Fig. 3.

12. The greater longevity of the 161 ppm peak suggests that it does not evolve into the larger PNS (169 ppm peak), and evolves directly into precipitated CaC, contrary to what the authors suggest.

Reviewer #4 (Remarks to the Author):

The manuscript, "Characterization of Early-Stage Pre-Nucleation Species (PNS) in Calcium Carbonate Crystallization," describe early stages of calcium carbonate formation. Utilizing nuclear magnetic resonance (NMR) spectroscopy enhanced by 'Bullet' dynamic nuclear polarization (Bullet-DNP), the study demonstrates the presence of transient PNS with lifetimes shorter than 5 seconds.

The authors evaluate the characteristic decay rates (R_{DEC}) and build-up rates (R_{BU}) observed in the D-DNP experiments, ensuring line widths are determined post-stabilization to avoid biases from shim instabilities. Despite these contributions, the manuscript requires several major revisions to improve clarity and depth.

- The current description of the non-classical crystallization pathway of calcium carbonate precursors under strong oversaturation is insufficiently detailed. Provide a more detailed explanation of the proposed non-classical crystallization pathway. Include a diagram or schematic that clearly illustrates the stages and mechanisms involved in the formation of calcium carbonate precursors under conditions of strong oversaturation. This will help readers better understand the novel aspects of the crystallization process being investigated.
- The results shown in Figure 2 lack comprehensive discussion, particularly concerning the classification of observed PNS. Discuss the results shown in Figure 2 in greater detail. Specifically, explain whether the observed PNS are (i) type 1 PNS or (ii) type 2 PNS, providing the reasoning behind this classification. This discussion should include a comparison of the observed characteristics of the PNS with established criteria for type 1 and type 2 PNS, as well as any relevant experimental data or theoretical considerations.
- The manuscript does not estimate the radius of the observed PNS, which is crucial for understanding their morphological characteristics. Provide an estimation of the radius of the PNS observed. This can be done theoretically, using established models and equations, or morphologically, if possible.
- Although the manuscript references a previous study on calcium phosphate nucleation using hyperpolarized real-time NMR, it would benefit from a more explicit comparison. Highlight similarities and differences in findings, methodologies, and implications between the two studies.
- Expand the discussion on the potential impact of these findings on the broader field of crystallization research. Outline possible future research directions that could stem from this study, particularly in exploring unexplored parts of the phase space for carbonate-based material formation.

Reviewer #5 (Remarks to the Author):

I co-reviewed this manuscript with one of the reviewers who provided the listed reports. This is part of a Communications Chemistry initiative to facilitate training in peer review and to provide appropriate recognition for Early Career Researchers who co-review manuscripts.

Reviewer #1 (Remarks to the Author):

Authors of the manuscript "Bullet-DNP enables in-situ observation of multiple short-lived calcium carbonate precursors – non-classical crystallization under strong oversaturation" described the process of in situ observation of multiple short-lived precursors of calcium carbonate–non-classical crystallization under strong supersaturation. Turhan et al. demonstrates how Bullet-DNP was used to investigate the initial stages of calcium carbonate precipitation under high supersaturation conditions. The authors further demonstrate how Bullet-DNP, which involves the transfer of a hyperpolarized frozen sample directly into the NMR spectrometer, makes it possible to detect several short-lived CaCO₃ PNS, whose lifetime is less than 5 seconds.

The manuscript is well structured and interesting to read. Here are some optional comments to address:

- What are the next steps to further elucidate the atomic-level structure of the short-lived PNS using Bullet-DNP in combination with other structural techniques?

We have added a paragraph to the manuscript commenting on possible combinations with other techniques. It reads:

“Considering that Bullet-DNP is sensitive to very early-stage precursors of CaC at relatively high ion strengths, a combination with molecular dynamics (MD) simulations appears as a possible future integration into established investigative frameworks in materials research. Not only are MD simulations also applicable to processes taking place on the microseconds time scale (or milliseconds when coarse-graining), but the high concentrations help to reduce computational costs, too, as box sizes can be reduced.”

- How does the degree of supersaturation quantitatively affect the lifetimes of the different PNS observed? and sizes of PNS?

This question is not possible to answer at present, as too many factors (such as co-solutes, temperature, pH) influence the lifetime of a PNS as well as its size. Therefore, we would like to refrain from giving a quantitative statement. However, qualitatively, it can be stated that higher degrees of oversaturation entail shorter lifetimes, as well as larger observed species. To take the referee’s point into account, we have added the following paragraph to the manuscript:

“Qualitatively, it can be stated that higher degrees of oversaturation entail shorter lifetimes, as well as larger observed species as reflected in broader lines¹. However, it should be noted that the faster kinetics at high concentrations and the associated acceleration of the solidification event also shift the fraction of the precipitation event, which Bullet-DNP observes.”

Reviewer #2 (Remarks to the Author):

Meier, Kurzbach and coworkers report an interesting application of “bullet dDNP”, a hyperpolarising liquid NMR technique developed a few years ago, to the investigation of the early stages of the formation of calcium carbonate. This application brings an important advantage as it allows to overcome current limitations of traditional dDNP approaches for the study of pressure sensitive samples, so it has a potentially large application to study fast events in these systems. The manuscript is concise and clear, the references appropriate, and the results are potentially interesting.

We thank the referee for his motivating words.

However this study lacks strong evidence for their conclusions, as I see a problem with the NMR data interpretation.

My main criticism on this study is related to the assignment of the two signals observed at 162 and 169 ppm (shown in figure 2) to clusters of different sizes.

In the manuscript, the signal with smaller linewidth (at 162 ppm) is assigned to smaller calcium carbonate clusters, while the one with the larger linewidth (169 ppm) to larger calcium carbonate clusters. While this interpretation is purely based on dynamics (larger clusters should be slower than smaller ones, which is understandable), I was surprised that isotropic chemical shifts are not discussed at all in the manuscript. In particular, it seems important to discuss why two carbonate species that only differ in their size should show such a large (more than 6 ppm) chemical shift difference?

We thank the referee for pointing this out. Indeed, we never argued that the local atomistic structures of the larger and smaller clusters are similar. However, we tried to avoid speculative statements as the local pH and exchange kinetics within the PNS are unknown and, thus, omitted analysis of chemical shift differences. We now emphasize in the revised manuscript that both the chemical environment and the size may be different for the two species:

“It should be noted that the chemical shift difference between the smaller and the larger PNS is >6 ppm. Hence, the two species also very likely differ in local structures (specifically the chemical environment of the ^{13}C nuclei), to which the chemical shift is sensitive. For example, local pH, ion coordination, hydration states, and exchange kinetics between different sites might differ. However, these factors are beyond the scope of Bullet-DNP. Hence, definitive statements about the structures of the PNS cannot be made at this point.”

It is claimed that this interpretation aligns well with previous articles on the formation of prenucleation clusters of calcium carbonate, with a citation to ref. 18 (Demichelis et al.). However, an important aspect of Ref 18 is the discussion of the role of bicarbonate ions in carbonate cluster speciation, at different pH. Now, in the present manuscript, there is no mention at all of the possible presence of bicarbonate ions. The (computational) study in Ref 18 was recently used to support interpretation of NMR data reported in another article from some of the same authors (cited as ref 22 in this manuscript). Ref 22 clearly shows that ^{13}C signals of bicarbonate ions in solid amorphous calcium carbonates appear at ~162 ppm, while signals of calcium carbonate in amorphous environments are expected at ~168 ppm.

Our statement referred to the fact that our data points toward the possible existence of several PNS. We now realize that this was too briefly sketched in the original submission. We have now extended the sentence in question and also included more references:

“However, other possible scenarios cannot be excluded, such as the species resonating at 161 ppm, does not transform into CaC, and free carbonate directly transforms into PNS without any intermediate step, and the second species forms without taking part in any further event. As the initial concentrations are much higher than after a few seconds, it is also possible that the system wanders from one part of the phase space to another. However, the reported POC data alone do not allow for the definitive pathway to be determined, yet our observation can be reconciled with the existing models, which show that CaC precipitation can cover several precursor species.²⁻⁴”

The reason for not mentioning bicarbonate in the present manuscript is that we only show liquid-state data. In the liquid state, bicarbonate does not exist as a distinct species (see Fig. R1). In dependence on the pH different equilibria with H_2CO_3 and CO_3^{2-} are formed. As a result, the chemical shift is also an exchange-averaged value. The referee’s assumption would require that two carbonate species exist in the slow exchange next in an aqueous solution (note that reference 22 is a solid-state NMR study). However, to the best of our knowledge, this is impossible for acids such as H_2CO_3 where proton exchange rates are typically in the GHz regime.

Figure R1. pH-dependence of relative populations of H_2CO_3 , HCO_3^- , and CO_3^{2-} (DOI: 10.5004/dwt.2009.700)

We have also measured references of our solutions to make sure that the signal assignment of the free carbonate species is correct (SI Fig. S6). To make this reasoning clear to the reader, we have now added the following statement to the main text:

“Note that bicarbonate species in solid amorphous CaC also display a chemical shift of 162 ppm. Hence, the PNS species at 161.4 ppm may also be primarily bicarbonate-based. However, the exchange kinetics between H_2CO_3 , HCO_3^- , and CO_3^{2-} are unknown, as are their relative populations within the PNS. Hence, definitive statements cannot be made.”

And later:

“[...] for the assignment of the chemical shift of the free species, see Fig. S6.”

Therefore, I strongly suspect that the assignment discussed in this manuscript is not correct. It is important that data interpretation is revised in light of the cited studies before the manuscript becomes acceptable for publication in this Journal.

Concluding our above answers, we have to (partially) disagree with the referee’s suspicion. The chemical shifts of solid CaC do not reflect the exchange-averaged values in solution-state NMR.

Further, our reference experiments clearly prove that the free carbonate species is correctly assigned (Fig. S6) and that under our conditions, there is no slow exchange between HCO_3^- and CO_3^{2-} , which would lead to two separate signals. To avoid any speculative structure assignments, consequently, all species not found in the reference experiments were deliberately generically denoted as PNS.

Furthermore, our assignment of free carbonate and precipitating PNS-based species in the Bullet-DNP experiments can also be verified by the different effective R_1 rates, which show that the species the referee suspects as bicarbonate also disappears from the solution at almost the same rate as the larger PNS. However, as stated in our response above, the smaller observed PNS may be based on calcium bicarbonate (in line with the referee's argument). Hence, this is now considered in the revised manuscript's new paragraphs.

Related to this point: concerning the discussion on the decay curves at different pHs ("higher pH led to faster conversion of the small ionic clusters into larger aggregates/condensed phases"), the observed behaviour could be justified, too, by considering the different assignment of the two observed peaks to calcium carbonate in prenucleation species (169 ppm) and bicarbonate ions (162 ppm), as, with increasing the pH, the equilibrium between bicarbonate and carbonate ions should be displaced more towards the carbonate ions.

Apologies if we overlook something obvious here, but this argument would also require a system in slow exchange (on the NMR timescale) between bicarbonate and another carbonate species in solution. However, given that the proton exchange rates of oxyanion-based acids in aqueous solutions are typically in the order of GHz, such an argumentation appears impossible to us. However, in solid-state NMR, the referee would be absolutely right. Hence, to take the referee's comment into account, we have added the following:

"The observation of the species at 161.4 ppm and at 160.4 ppm through to distinct resonances necessitates slow exchange between these two, which renders yet it unlikely that the resonance at 161.4 ppm is free bicarbonate in solution."

Other minor points:

-The manuscript claims that that standard dDNP techniques are incompatible with the detection of calcium carbonate clusters because of the degassing step. However, no proof is provided or previous reference. It would be instructive to see these data added to the SI.

These data only show the glycerol signals. We have added these to the revised Supporting Information.

-It is not clear why such high supersaturation levels were employed in this study. Why are these conditions relevant? This point should be mentioned in the revised manuscript.

We have added this information to the manuscript. These were employed to probe unstudied parts of the phase space so far, which are well compatible with experimental dissolution DNP conditions.

Reviewer #3 (Remarks to the Author):

The work of Turhan et al. uses a specific and novel form of hyperpolarized solution NMR spectroscopy where the sample is shuttled rapidly from the hyperpolarizer to the NMR magnet, where it is dissolved for study. In this case, this enables the signal-enhanced time-resolved study of product nucleation during a solution reaction between sodium carbonate and calcium chloride. Interestingly, different carbonate species are observed after only a few seconds of mixing. This is clearly an important technique that will

help researchers understand the first steps of crystallisation. However, there are some remaining questions that need to be addressed before this work can be published. These are provided below:

1. What polymorph(s) of CaC are precipitated from this reaction? The pXRD of the precipitate should be shown. Do the PNS lead to one polymorph, or two (or more)? Does this vary with pH? If so, can this be linked to the particular PNS as the larger clusters are more prevalent at higher pH?

We performed the pXRD experiments, as the referee suggested. These showed a co-existence between vaterite and calcite, which shifted towards the latter with decreasing pH.

pH-6: 83 % Vaterite (32 nm)
17 % Calcite (207 nm)

pH-7: 91 % Vaterite (30 nm)
9 % Calcite (205 nm)

pH-8: 97 % Vaterite (31 nm)
3 % Calcite (95 nm)

However, given that both CaC types coexist at all probed concentrations, no straightforward correlation with the cluster size can be drawn. We have added this information to the revised Supporting Information and included a reference to it in the main text:

“Upon completion of the solidification process, the solid powders were analyzed by powder X-ray diffraction, yielding different contributions of vaterite and calcite in dependence of the pH (see Supporting Information). Hence, the observed PNS did not lead to one specific solid-making function correlations complicated.”

2. What role does glycerol play in NC formation? This should be discussed in the main text and not only in the SI. In the SI the authors state that glycerol can be assumed to not take part in CaC formation, with a citation, and yet the cited paper (by some of the authors) states: “It should be mentioned that the D-DNP approach employs additives such as glycerol and the polarizing agent (TEMPOL). These specific conditions might influence the precipitation process, which therefore cannot directly be compared to previously published data on CaP systems.” So there is a contradiction. It could well be expected that glycerol plays a role in stabilizing clusters as a ligand, as water would also be expected to do. The experiments should be repeated using DMSO (or equivalent cryoprotectant) to ensure the same species are observed. The sodium carbonate may help pure water form a glass and thus no cryoprotectant may be required for DNP. It is worth trying.

We attempted the reference experiments the referee suggested. However, the solubility of carbonate in water/DMSO mixtures was such that the carbonate always precipitated upon freezing the bullet. Therefore, it was impossible to apply cryogenic DNP. Similarly, no DNP was observed when all cryoprotectants were avoided together. Nevertheless, the referee’s comment is valid, and we have therefore included the following statement in the SI to the description of the correction procedure:

“However, the presence of glycerol may have an impact on the kinetics and dynamic of CaC formation.”

In the main text, we added the following statement:

“The reported kinetic data potentially remain biased to some degree by concentration gradients as the reaction rate might change. The presence of glycerol may likewise impact the CaC formation.”

3. State the full name of HEPES.

We do this now.

4. Fig. S6 missing.

Our apologies: this was overlooked during formatting (a spurious figure caption). The figures and order of the SI have now been fixed.

5. What's happening in Fig. S1? There is not sufficient information in the caption to understand where this exemplary spectrum comes from – and it's labelled as exemplary but does not resemble others (e.g. Fig. 2). Why is the peak near 169 ppm so much larger than in Fig. 2?

We have added more information to the figure caption. The spectrum was recorded at pH 7 immediately after the start of the detection period. Thus, the differences between the figure are due to the differences in pH.

6. Why is the noise larger at shorter times (Fig. 2, S3 (left), but not S3 (right))? Yet it's the same in S4. This should be explained.

This is due to the applied correction procedure. As sample convection can reduce the signal intensities, correcting the data as outlined in the Supporting Information leads to an artificial increase in noise, particularly shortly after the start of the detection period. We now realize that the original note: "Note that the noise level is changing because of the data correction procedure (see the Supporting Information)." was too brief to explain this circumstance clearly. We have, therefore, added another paragraph to the main text. It reads:

"Finally, note that the sample convection can reduce the signal intensities. Hence, correcting the data as outlined in the Supporting Information leads to an artificial increase in noise, particularly shortly after the start of the detection period (see, e.g., Fig. 2). However, under strong convection in combination with possible sample inhomogeneity, the substrate hyperpolarization cannot be derived directly from the signal-to-noise ratio."

7. Fig. 3 does not look like it matches the data. Also, the caption should specify which pH sample. Are they relative to carbonate intensity? If so, this wouldn't be S/S₀.

How is S/S₀ above 1?

We now report the pH explicitly in the figure caption and explain that S/S₀ means that the data is normalized to the first data point shown in the figure, *i.e.*, to a maximum value of 1. The visual mismatch in the referee remarks may be due to this normalization procedure. Numerically, all data are sound after doublechecking. To avoid any further confusion, we have now added the non-normalized data to the SI of the revised manuscript (see also our subsequent response).

This should be clarified in the caption. For the 169 ppm resonance, it is almost gone at 3 s (as stated below in the text), whereas the 161 ppm resonance remains and appears to have a different decay constant in Fig. 2, whereas in Fig. 3 it appears the same for the 169 and 161 species. Something is amiss.

Indeed, the chosen representation (with the normalization to 1) can be visually misleading. In fact, the strong signal amplitude at $t = 0$ leads to the impression that the signal decays to naught. We have, therefore,

added the non-normalized data (Fig. R2) to the Supporting Information. We refer to the added data in the updated figure caption of Fig. 3.

Figure R2. Non-normalized signal intensities

The difference in decay between the two PNS species is not as pronounced as the difference between PNS and free carbonate, but it can be clearly derived from the fits to the data (dotted lines in Fig R2).

8. Main text has integral ratios measured after 1 s, whereas caption to Fig. S7 has 2 s. Also, integral ratios should not be measured in %. An integral "ratio" of 96% could suggest a ~1:1 ratio.

We thank the referee for catching this. There is a difference between the start of the detection period and the mixing of the two samples. We now realized that this was confusing in the original submission. We have made this clear in the revised version.

Further, we have updated the ratio notation as the referee suggested.

9. Why narrower for the 169 ppm peak at pH 7? Same species but small cluster? Does this infer there are no magic cluster sizes here?

The referee is correct; the PNS size, as well as the observed part of the precipitation pathways, may be pH-dependent. This also means that the cluster size (either 'magic' as in C60 or not) can also be pH-dependent. We have added a corresponding comment to the manuscript to point toward this circumstance. It reads:

"Note that the line widths of the resonances at 169 ppm are changing with pH. This might be due to varying PNS sizes with pH, as well as changing growth kinetics. Hence, our experiments suggest that not a single PNS size can account for all probed conditions, which is well in line with the reported literature.^{1,2,4-7}"

10. Why is the black spectrum in Fig. 4 different to the blue spectrum in Fig. S7 when they have been recorded under the same conditions? Should they not be the same spectrum? How many repeat measurements were done? I think in the main text is pH 6, that's why. What is then the error on the integral ratios? It looks very large comparing Figs. 4 and S7 for the pH 6 sample.

The difference is due to the fact that these two spectra were taken at different time points (2 s after the start of the detection period or directly after mixing). We now realized that this was unclear in the original submission and have updated the text/figures accordingly. But the data set is the same.

We have also now added the errors of the integral ratios.

11. Typo: “However, it must be considered that the kinetic data shown in Fig. 4 potentially remains” should be Fig. 3.

We have corrected this.

12. The greater longevity of the 161 ppm peak suggests that it does not evolve into the larger PNS (169 ppm peak), and evolves directly into precipitated CaC, contrary to what the authors suggest.

If we understand the referee correctly, the remaining signal at 161 ppm, which does not transform into CaC means that free carbonate directly transforms into PNS without any intermediate step and that the second species forms without taking part in any further event.

After reflecting on this argument, we agree with the referee that this is not impossible. However, our original interpretation would still be reconcilable with such an argument as the initial concentrations are much higher, such that the suggested process takes place on shorter time scales until the system drops to reduced saturation conditions, which stabilize smaller PNS.

However, the reported POC data alone do not allow for the definitive pathway to be determined. Hence, we have added the referee’s argument to the main text and rephrased it accordingly so that other possible scenarios cannot be excluded:

“However, other possible scenarios cannot be excluded, such as the species resonating at 161 ppm, does not transform into CaC, and free carbonate directly transforms into PNS without any intermediate step, and the second species forms without taking part in any further event. As the initial concentrations are much higher than after a few seconds, it is also possible that the system wanders from one part of the phase space to another. However, the reported POC data alone do not allow for the definitive pathway to be determined, yet our observation can be reconciled with the existing models, which show that CaC precipitation can cover several precursor species.²⁻⁴”

Reviewer #4 (Remarks to the Author):

The manuscript, "Characterization of Early-Stage Pre-Nucleation Species (PNS) in Calcium Carbonate Crystallization," describe early stages of calcium carbonate formation. Utilizing nuclear magnetic resonance (NMR) spectroscopy enhanced by 'Bullet' dynamic nuclear polarization (Bullet-DNP), the study demonstrates the presence of transient PNS with lifetimes shorter than 5 seconds.

The authors evaluate the characteristic decay rates (R_{DEC}) and build-up rates (R_{BU}) observed in the D-DNP experiments, ensuring line widths are determined post-stabilization to avoid biases from shim instabilities. Despite these contributions, the manuscript requires several major revisions to improve clarity and depth.

- The current description of the non-classical crystallization pathway of calcium carbonate precursors under strong oversaturation is insufficiently detailed. Provide a more detailed explanation of the proposed non-classical crystallization pathway. Include a diagram or schematic that clearly illustrates the stages and mechanisms involved in the formation of calcium carbonate precursors under conditions of strong oversaturation. This will help readers better understand the novel aspects of the crystallization process being investigated.

We have added another paragraph to the manuscript to take the referee's point into account and to discuss the differences between non-classical crystallization and nucleation-and-growth. However, we want to refrain from making statements about the theoretical classification of the observed species (as already noted in the original submission's introduction) as we propose a methodology that can be used in either context. Hence, the added paragraph reads:

"PNC are thought of as (meta-)stable ionic assemblies, which form in solution and represent the first step in the formation pathway of many solid ceramics. Although different PNC properties under different experimental conditions have been reported, many observations report a coalescence and dehydration of PNC that lead to the formation of solid phases. Such solidification mechanisms are challenging the classical nucleation-and-growth theory (CNT). Yet, it is still debated whether the observation of PNC can be reconciled with the CNT.^{4,5,8} The discussion is further complicated by the fact that experimental evidence is scarce, not least due to the highly dynamic character of PNC."

However, we want to refrain from introducing a diagram showing the precipitation pathway, in line with Referee 4's argument that the proposed pathway might involve further features that remain undisclosed using the currently available data set. A detailed mechanistic description would go beyond the scope of the reported methodological proof-of-concept, and a diagram might give a misleading impression of a definitive mechanism.

- The results shown in Figure 2 lack comprehensive discussion, particularly concerning the classification of observed PNS. Discuss the results shown in Figure 2 in greater detail. Specifically, explain whether the observed PNS are (i) type 1 PNS or (ii) type 2 PNS, providing the reasoning behind this classification. This discussion should include a comparison of the observed characteristics of the PNS with established criteria for type 1 and type 2 PNS, as well as any relevant experimental data or theoretical considerations.

We followed the referee's suggestion and added more details on the PNS characterization. The added paragraph reads:

"Earlier work classified PNS into type 1 and 2 PNS.^{1,9} The former are stable (or metastable) in solution and only initiate the solidification process upon external stimuli, such as pH or temperature jumps. The latter

appears as a transient species in precipitation events, which reach the final solid without any further trigger (although this definition does not exclude longer steady-state phases). For the PNS reported herein, evidently, the latter case applies, *i.e.*, the observed PNS are immediately transformed into solid without the need for any further stimuli given the high degrees of oversaturation.”

Again, we want to refrain from classifying the observed species into the models, which are still under discussion in the literature, as the proposed methodology can be applied in either case. Due to this, we also describe in the manuscript that the notion PNS is “thought of as a generic description of material precursor species, not invoking any particular definitions.”

- The manuscript does not estimate the radius of the observed PNS, which is crucial for understanding their morphological characteristics. Provide an estimation of the radius of the PNS observed. This can be done theoretically, using established models and equations, or morphologically, if possible.

We followed the referee’s suggestion. Considering that the hydrodynamic radius of bicarbonate¹⁰ is ca. 0.2 nm and based on the framework established in reference 1, an increase in linewidth by a factor of $318/9 = 35.3$ corresponds to an at a most 35-fold increase in rotational correlation time and at most 3.3-fold increase in hydrodynamic radius. We have added this information to the manuscript, keeping in mind that chemical exchange is neglected as a possible bias. The new paragraph reads:

“Considering that the hydrodynamic radius of bicarbonate¹⁰ is ca. 0.2 nm and following the framework established in reference 1, an increase in linewidth by a factor of $318/9 = 35.3$ corresponds to an at a most 35-fold increase in rotational correlation time and, thus, a <3.3-fold increase in hydrodynamic radius. However, it should be noted that the linewidth Γ_2 can also be significantly influenced by exchange processes, such that the 3.3-fold increase must be considered as a theoretical maximum.”

- Although the manuscript references a previous study on calcium phosphate nucleation using hyperpolarized real-time NMR, it would benefit from a more explicit comparison. Highlight similarities and differences in findings, methodologies, and implications between the two studies.

Again, we followed the referee’s suggestion and added the following paragraph to the conclusion of the manuscript:

“It should be noted that earlier work on calcium phosphates using dDNP to monitor prenucleation species led to comparable observations, *i.e.*, concerted detection of PNS and free phosphate ions in solution. However, as calcium phosphates feature a much lower solubility than carbonates these studies were performed at lower degrees of oversaturation. More importantly, though, the main difference was that ‘traditional’ dissolution DNP could be applied as phosphate ions remain in solution even under high chase gas pressures. In combination, though, DDNP and Bullet-DNP provide means to study prenucleation phenomena for a wide range of target systems, including biologically most important oxyanions.”

- Expand the discussion on the potential impact of these findings on the broader field of crystallization research. Outline possible future research directions that could stem from this study, particularly in exploring unexplored parts of the phase space for carbonate-based material formation.

Taking the referee’s comment into account while at the same time trying not to become too speculative, we have now added the following paragraph to the conclusion of our manuscript:

“The reported methodology can potentially help to widen the accessible precursor space in the design of CaC-based functional materials. In particular, endeavors that aim to predetermine the morphology and structure of a carbonate-based solid by choice of a particular PNS might profit from the reported methodology. The

solidification pathways from a PNS to a final material in the almost untapped high oversaturation regime of the CaC phase space might potentially lead to new morphologies with unexpected properties.”

Reviewer #5 (Remarks to the Author):

I co-reviewed this manuscript with one of the reviewers who provided the listed reports. This is part of a Communications Chemistry initiative to facilitate training in peer review and to provide appropriate recognition for Early Career Researchers who co-review manuscripts.

We thank the referee for his contribution.

References

- 1 Weber, E. M. M. *et al.* Assessing the Onset of Calcium Phosphate Nucleation by Hyperpolarized Real-Time NMR. *Anal Chem* **92**, 7666-7673, doi:10.1021/acs.analchem.0c00516 (2020).
- 2 Demichelis, R., Raiteri, P., Gale, J. D., Quigley, D. & Gebauer, D. Stable prenucleation mineral clusters are liquid-like ionic polymers. *Nat Commun* **2**, doi:ARTN 590 10.1038/ncomms1604 (2011).
- 3 Ramnarain, V. *et al.* Monitoring of CaCO₃ nanoscale structuration through real-time liquid phase transmission electron microscopy and hyperpolarized NMR. *Journal of the American Chemical Society* **144**, 15236-15251 (2022).
- 4 Avaro, J., Moon, E. M., Schulz, K. G. & Rose, A. L. Calcium Carbonate Prenucleation Cluster Pathway Observed via In Situ Small-Angle X-ray Scattering. *J Phys Chem Lett* **14**, 4517-4523, doi:10.1021/acs.jpcclett.2c03192 (2023).
- 5 Henzler, K. *et al.* Supersaturated calcium carbonate solutions are classical. *Sci Adv* **4**, eaao6283, doi:10.1126/sciadv.aao6283 (2018).
- 6 Mohammed, A. S. A., Carino, A., Testino, A., Andalibi, M. R. & Cervellino, A. In Situ Liquid SAXS Studies on the Early Stage of Calcium Carbonate Formation. *Part Part Syst Char* **36**, doi:ARTN 180048210.1002/ppsc.201800482 (2019).
- 7 Huang, Y. C. *et al.* Uncovering the Role of Bicarbonate in Calcium Carbonate Formation at Near-Neutral pH. *Angewandte Chemie International Edition* **60**, 16707-16713 (2021).
- 8 Carino, A., Ludwig, C., Cervellino, A., Muller, E. & Testino, A. Formation and transformation of calcium phosphate phases under biologically relevant conditions: Experiments and modelling. *Acta Biomater* **74**, 478-488, doi:10.1016/j.actbio.2018.05.027 (2018).
- 9 Zahn, D. Thermodynamics and Kinetics of Prenucleation Clusters, Classical and Non-Classical Nucleation. *Chemphyschem* **16**, 2069-2075, doi:10.1002/cphc.201500231 (2015).
- 10 Kadhim, M. & Gamaj, M. I. Estimation of the diffusion coefficient and hydrodynamic radius (stokes radius) for inorganic ions in solution depending on molar conductivity as electro-analytical technique-a review. *J. Chem. Rev* **2**, 182-188 (2020).

REVIEWERS' COMMENTS:

Reviewer #1 (Remarks to the Author):

Authors have properly addressed the comments. I recommend to accept this manuscript.

Reviewer #2 (Remarks to the Author):

These authors responded to all the points, and manuscript can now be accepted. I congratulate the authors on the highly improved quality of the discussion.

Reviewer #3 (Remarks to the Author):

Substantial changes have been made to the manuscript and it is now vastly improved. While the paper is lacking in some important conclusions, such as the structures present in the PNS, this was not the aim of the work and I can understand (and appreciate) the authors' reluctance to make bold claims. Being able to detect these species and highlight that they change (if not how they change) with respect to environmental conditions is important. There are a lot of questions remaining, but hopefully they can be answered in with further studies. I believe the work is now publishable.

Reviewer #4 (Remarks to the Author):

Thanks to the authors: they successfully addressed my concerns. I recommend that the manuscript be accepted for publication.

Reviewer #5 (Remarks to the Author):

I co-reviewed this manuscript with one of the reviewers who provided the listed reports. This is part of a Communications Chemistry initiative to facilitate training in peer review and to provide appropriate recognition for Early Career Researchers who co-review manuscripts.